# Understanding the neurodynamic process of decision-making for mobile application downloading

**Harshit Parmar[1]**, **Fred Davis[2]**, **Eric Walden[2]** *

**1** Texas Tech Neuroimaging Institute, Texas Tech University, Lubbock, Texas, United States of America,
**2** Rawls College of Business, Texas Tech University, Lubbock, Texas, United States of America

☯ These authors contributed equally to this work.
* eric.walden@ttu.edu

## Abstract

In this article, we try to explore and understand the neurodynamics of the decision-making process for mobile application downloading. We begin the model development in a rather unorthodox fashion. Patterns of brain activation regions are identified, across participants, at different time instance of the decision-making process. Region-wise activation knowledge from previous studies is used to put together the entire process model like a cognitive jigsaw puzzle. We find that there are indeed a common dynamic set of activation patterns that are consistent across people and apps. That is to say that not only are there consistent patterns of activation there is a consistent change from one pattern to another across time as people make the app adoption decision. Moreover, this pattern is clearly different for decisions that end in adoption than for decisions that end with no adoption.

## Introduction

Decision making is a complex process which has been well studied for decades. Most of the earlier studies on decision making are socially and psychologically inclined. With the advent of functional neuroimaging, it was made possible to understand the physiological aspect of decision making. Numerous studies have revealed different factors affecting the decision-making ability such as reward [1, 2], risk [3, 4], uncertainty [5, 6], morality [7, 8], etc. These studies also identify discrete brain regions for quantifying abstract features like risk, morality, uncertainty, etc. However, the majority of these studies are focused on a single aspect of the decision-making rather than the entire process [9]. In the earlier days of cognitive neuroscience, more priority was given in identification of brain regions responsible for different aspects. Currently, with decent knowledge on the functionality of major brain regions focus should be shifted to analyze the decision-making process as a whole rather than bits and pieces. Attempts has been made to study the process of decision making, but the process model is more of a theoretical framework [10, 11].

In this paper, we investigate the decision-making process from neuroscience point of view using functional Magnetic Resonance Imaging (fMRI). This study is more of an explorative

**Data Availability Statement:** The entire dataset has been uploaded to 'Texas Data Repository'. The link to the dataset is as follows: https://doi.org/10.18738/T8/QICUQB.

**Funding:** Funding was provided to FD by the Rawls College of Business. The funders had no role in study design, data collection and analysis, decision to publish, or preparation of the manuscript.

**Competing interests:** The authors have declared that no competing interests exist.

approach aimed to identify various stages in the decision-making process. We track the entire decision-making process inside the brain. The decision-making process was studied using a pseudo-realistic mobile application download scenario. The mobile app adoption scenario is chosen for 2 main reasons. First, the app adoption is closely related to technology acceptance, a field studied well for decades. There are many models focusing on understanding the adoption of information technology in general. The technology acceptance model (TAM) introduced more than 30 years ago [12] remains the leading behavioral model for predicting, explaining, and increasing user acceptance. There have been numerous extensions of this model over the years [TAM2: 13; TAM3: 14; UTAUT: 15]. But all of these models are attempts to explain information technology adoption in the way it was done when the TAM model was developed. There are fMRI studies which support the TAM model for decision-making process [16] and might act as a starting point towards development of a psycho-cognitive model for decision making process.

Second, the impact of app adoption on daily basis. In 2019 consumers spent $83.5 billion dollars on apps on Google Play and in the Apple App store [17]. This is more than the 2019 revenue of Oracle, SAP, and Salesforce combined! This is almost twice as much as worldwide spending on database management systems [18]. Moreover, the total revenues generated from the Apple app store alone including retail purchases through apps, ride hailing, streaming and app-based advertising was $519 billion [19]. With the tremendous amount of money being spent on apps and the even more prodigious amount being generated through the use of apps, it is worthwhile to develop a comprehensive model of how app adoption decisions are made.

Describing the process of how the decision unfolds over time in the brain is useful for several reasons. A neural process model is an end in and of itself, which can be used in the same way a static self-report model is to make predictions and inform decisions, though it may require specialized equipment to use. A process model provides details that a static model does not provide, which can create new insights for researchers and practitioners on how to facilitate adoption. A neural process model can combine with self-reported static models to increase prediction accuracy. Whereas a static model based on self-reports can only discover what the subject knows and what the researcher asks, a neural model has the potential to capture process that the decision maker was not aware of and/or processes that researchers had not thought to explore.

## Methods

Written consent was obtained from all the participants prior to commencing the experiment. All the participants were also screened for MRI (Magnetic Resonance Imaging) safety with a separate MRI safety screening form. This study was approved by the Texas Tech Human Research Protection Program with number IRB2016-557.

### Experiment protocol

For this exploration we downloaded descriptions of 25 apps from PopularMechanics.com, which publishes many lists of app recommendations. These recommendations have a screenshot and short textual description. Thus, there is ecological validity in that these are real descriptions from a real website. However, these are not the descriptions in the app store, which would be too large and complicated to display in an MRI machine. Participants were shown the information about the app and asked to indicate if they would download an app.

While in the MRI scanner, anatomical brain scans were performed first followed by the functional scans. The app stimulus was displayed on the screen. The flow of the experiment inside the scanner is illustrated in Fig 1. At the end of the anatomical scans, the participants

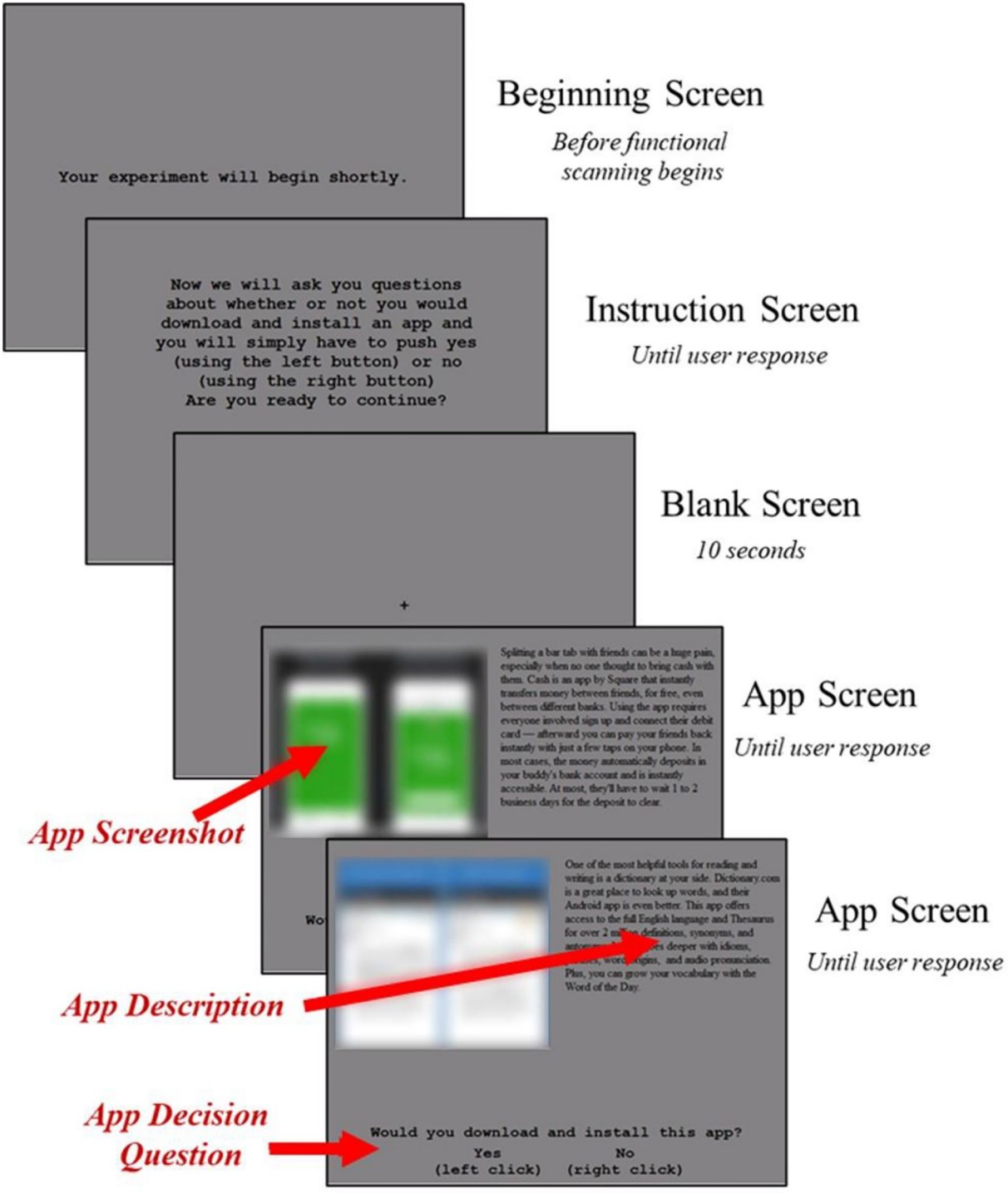

**Fig 1. Example stimuli screen.**

were informed about the beginning of the functional run. During this time, the 'Beginning Screen' would appear. Once the functional scans start, the 'Instruction Screen' is presented to the participants. The instruction screen contains a brief description about the task to be performed. The participates then indicate with a button press if they wish to continue further or

not. Once the participant decides to continue with the experiment, there is a 10 second 'Blank Screen'. The blank screen acts as a starting point and allows the blood oxygenation to reach a baseline value. After the blank screen the description of the app is shown in the 'App Screen'. There are three main parts of the app screen. First, the screenshot of the app, second the textual description of the app and third the question to download or not. Participants were allowed as long as they wanted to evaluate the app. When they had reached a decision subjects indicated whether or not they would download the app. Once the decision is made, app screen for a different app is presented. This sequence of events continue till the participants have responded to all 25 apps.

As soon as the subject presses the button a new app appears on the screen, in order to avoid contamination with the default mode network that may occur with a delay. During an inter-stimulus interval it is not clear what subject's brains would be doing. Presumably they would be preparing in some way to switch to the default mode network. As we are characterizing the time course of activation, rather than subtracting average activation in two different conditions, an inter stimulus interval would lead to extraneous activation. The trade-off is that activation at the end of stimuli N will bleed over into the beginning of stimuli N+1. Moreover, the experiment paradigm is kept to be more close to an actual app store browsing experience where there is no inter stimulus interval scrolling between two apps.

## Subjects

Subjects were 20 students from a large US university, who received course credit for participation. There were eleven males and nine females with an average age of 20.45 (st dev = 1.64).

## Scanner parameters

The dataset consists of MRI scan volumes for 20 subjects. All the MRI scans were performed using a 3T Siemens Skyra MRI machine. Anatomical scan was performed first using the sagittal MPRAGE pulse sequence. 192 sagittal slices were obtained with a slice thickness of 1 mm. The in-plane matrix size was 256 x 256 with a resolution of 1mm x 1mm. The functional scans were obtained using a multiband Echo Planar Imaging (EPI) technique. For functional scans, 48 axial slices were obtained with a slice thickness of 3.3mm. The in-plane matrix size was 82 x 82 with a resolution of 3.3mm x 3.3mm. The University of Minnesota Center for Magnetic Resonance Research EPI Multiband sequence [20] was used with an acceleration factor of six. Because of the use of multiband pulse sequence, the Repetition Time (TR) was 0.545 seconds. The total number of functional varied from subject to subject.

## fMRI analysis

**Preprocessing.** The fMRI dataset for all subjects were preprocessed before any analysis was performed. All the preprocessing was done using SPM12 (Statistical Parametric Mapping) toolbox [21] and some in-house scripts using MATLAB R2020a. Dataset for all the subjects went through the same preprocessing pipeline. The main preprocessing steps include motion correction, coregistration, normalization, spatial smoothing and temporal signal drift reduction. The motion artifacts were estimated and reduced using SPM12's 'Realign' option. All the functional volumes for a given subject, were aligned to the mean volume of the entire functional run. The alignments were performed using rigid body affine transform and 6 motion parameters, 3 translational and 3 rotational, were estimated and corrected for. Next, the functional volumes were coregistered to the anatomical scan volume. Coregistration was performed using the 'Coregister' option form SPM12. Then, the functional volumes were normalized to the standard MNI space using the 'Normalize' option from SPM12. Normalization

allows for comparison of results across different subjects. After normalization, functional data from all the subjects are in the same MNI space, 3 x 3 x 3 mm$^3$ voxels, and of same matrix size, [53 63 52]. After normalization, the functional volumes were smoothened using a 3D Gaussian filter with a Full Width Half Maxima (FWHM) of 8 mm. Finally, the temporal signal drift was removed using a PCA based detrend approach [22].

**GLM analysis.** To begin, a GLM (Generalized Linear Model) analysis was performed with an event related experiment design with event location corresponding to the button press. The events were assumed to be of 1 second in duration, beginning 1 second before the button press and ending with the button press. The choice of 1 second events was arbitrary. The canonical double gamma model for the hemodynamic response function (HRF) was used. Two separate regressors were used corresponding to the apps with 'Yes' and 'No' response. Three different contrasts were calculated namely: *'Yes > No'*, *'No > Yes'* and *'Both > Baseline'*. The first two contrasts identify regions that are involved more strongly in either Yes [+1–1] or only No [-1 +1] responses to the download question. The third contrast, both [+1 +1], corresponds to the brain response irrespective of the download decision. The GLM analysis was performed using the SPM12 toolbox [21].

**Functional connectivity analysis.** Functional connectivity analysis was used to first analyze the dynamic aspect of the data. Functional connectivity analysis is commonly used tool for analysis of resting state fMRI data [23] as there is no predefined experimental paradigm. The first step towards performing a functional connectivity analysis was to divide the entire brain into functionally similar Regions of Interest (ROI). Independent Components Analysis (ICA) algorithm was used for functional parcellation of the brain [24]. Group ICA analysis was performed on MATLAB 2020a using the GIFT ICA toolbox [25]. The ICA algorithm was set up to obtain 50 independent components and thus 50 functional ROIs. A representative time series for each ROI was obtained from the Independent Components (ICs) itself. Frequency analysis was performed on the representative time series and a fALFF score was computed. The fALFF score represents the low frequency to high frequency power ratio. The ICs with very low fALFF score ($<$1) were discarded from further analysis as those corresponded mostly to noise components. The spatial location for most those ROIs (with low fALFF score) were also in regions like the brain stem, eyes, and ventricles which suggest that no significant functional information can be extracted from those ICs. [Details about all 50 ICs and the corresponding fALFF scores is given in the appendix of S1 File].

In resting state fMRI data, dynamic Functional Connectivity (dFC) is usually computed using sliding window approach. In dFC analysis, pairwise correlation is computed between different region pairs using a section (window) of the entire time series. Multiple covariance matrices are computed for different sections of the time series. The changes in the covariance matrices for different time windows provides the temporal information. For resting state data, as there is no experiment paradigm, dFC analysis is performed on entire time series using time windows which overlap with one another. The location of the time window can be arbitrary for resting state data analysis.

In this case, the timing information about the beginning and the end of the stimulus is available thus the dFC time windows are selected accordingly. However, the response time for each app is different for different subject, thus a direct point to point comparison is not possible for time windows of varying length. To overcome this issue, a fixed window length of 10 seconds was used for all subjects. The response time for different apps for all 20 subjects is shown in Fig 2. It can be observed that for some subjects the response time for most of the apps was less than 10 seconds. Thus, only the subjects having a response time of greater than 10 seconds for at least 13 apps ($>$50% of the total apps) were considered for further analysis.

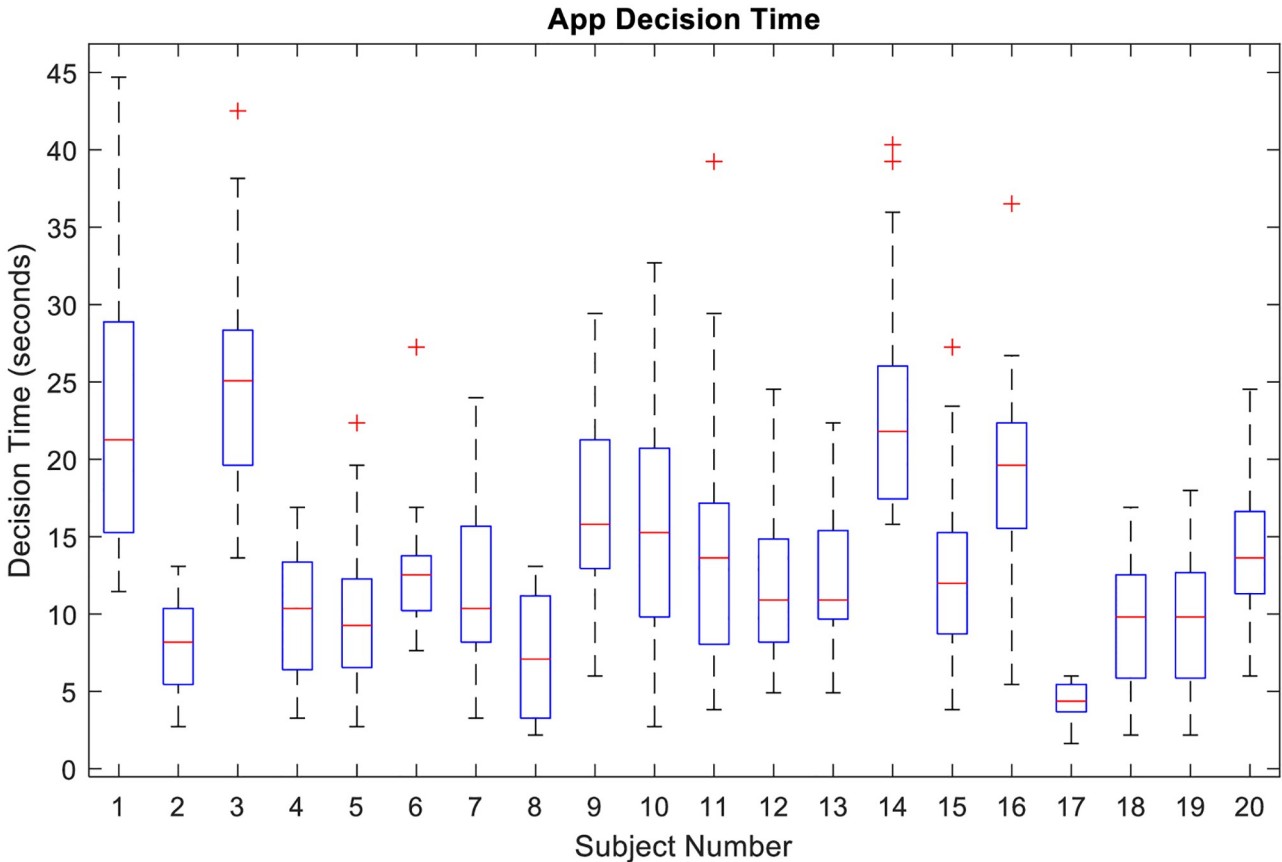

**Fig 2. Distribution of subject wise decision time for all 25 apps.** Each column shows the boxplot of the decision (response) time in seconds for the corresponding participant. The central red line indicates the median, the top and bottom edge of the box corresponds to the 75th and 25th percentile respectively and the outliers are indicated by the red '+' symbol.

The thresholding excluded 6 subjects. For the remaining 14 subjects, two separate dFC analyses were conducted. One where the time window starts at the start of the stimulus and other where the time window ends at the end of stimulus. The first case gives us information about the neurodynamics occurring at the beginning of the stimulus while the later about the ending of the stimulus. We assume that most of the information receiving and initial processing would occur in the beginning of the stimulus while the decision-making aspect would be highlighted more towards the end.

The dFC analysis was performed in two stages: subject level and group level. The subject level dFC analysis was conducted by first computing a pairwise correlation between time series of all ROIs for a single subject. Along with the correlation coefficient, the corresponding p-value was also computed. The correlation coefficient corresponding to a p-value of 0.05 or less (statistically significant correlation) were kept for further analysis while other (non-significant) were discarded. The thresholded correlation values for each pair of ROIs were averaged across all 25 apps which yields a subject level covariance matrix with values only for significant ROI pairs. Repeating the analysis for all the subjects gives us 14 separate covariance matrices. Along with the covariance matrices, a binary mask was also created to identify the pair of ROIs having significant similarity. For the group level analysis, the region pairs were further thresholded to only retain those region pairs which have significant correlation for more than half (>7) of

the subjects. Thus, key region pairs, consistent across subjects, showing high communication with one another, are identified.

Once the key region pairs are identified, the time series of those regions were analyzed to obtain useful temporal information. Some of the ICs, consists of more than one brain region spatially. The spatial regions for such ICs thus need to be subdivided into individual brain regions. The Automated Anatomical Labeling (AAL) brain atlas was used for masking the ICs [26–28]. Using the atlas, each IC was subdivided into multiple atlas regions. Corresponding to each brain atlas region, a subject-wise mean time series was obtained for different brain networks. Similar to dFC analysis, two separate time windows of 10 seconds each, were combined to obtain a time series which starts 10 seconds before the button press and ends 10 second into the new app after the image for new app is shown. The subject wise and the mean time series were analyzed to identify temporal trends for different regions.

**Spatiotemporal analysis.** The dFC analysis identifies key brain regions and the generalized trends in the time series of those regions. However, the absolute trends identified can be either localized towards the beginning or towards the end of stimulus. To identify the neurodynamics across the entire time of stimulus, comparison between windows of different time lengths is required. The comparison was achieved by normalizing different time windows to same lengths. Normalization was achieved by a two-step resampling process. The entire process is discussed later in this section.

The preprocessed fMRI data was first divided into different brain regions using the atlas. Different ROIs were obtained by masking the mean functional volume with AAL atlas. The atlas was first coregistered to the MNI space to ensure that the atlas has the same spatial dimensions as the functional volumes. The 'Coregistration: Estimate and Reslice' function of the SPM12 toolbox was used for coregistering the atlas to MNI space. The interpolation was set as nearest neighbor to avoid addition of intermediate values at the boundary of different ROIs. Moreover, a binary 3D mask was also obtained from the atlas that masked out the region outside the brain. The masking process reduces the effective number of voxels to be analyzed further.

The AAL atlas indexes 170 different potential ROIs. Detailed list of each ROI is given in the appendix section of S1 File. Each of the ROI was inspected manually and only the ROI having overlap with gray matter and cerebellum were considered for making with the functional volume as neuronal activation is only expected inside the gray matter region of the brain. [There is a debate in the neuroscience community about white matter activations but for this study only the grey matter activations were considered.] ROIs in the ventricles, brain stem, and outside the brain were discarded. After removing other ROIs, a total of 120 ROIs were masked to the functional volume. For each ROI, a representative time series was obtained. The representative time series was obtained by averaging the temporal BOLD (Blood Oxygenation Level Dependent) fluctuations of all the voxels inside a given ROI. Thus, we end up with 120 representative time series for each subject.

Again, a two-level analysis was performed to extract the spatiotemporal dynamics from the representative time series. The entire time series consists of the temporal fluctuations for all 25 apps. For subject level analysis of the temporal response of each app, the entire time series needs to be divided into 25 parts. Because the response time to each apps were different, a direct time point to time point analysis is difficult. Thus, the entire representative time series was first segmented into 25 parts corresponding to the timing for each app and then resampled to have a fixed number of time points. From Fig 2, it can be observed that the inter-subject variability in response time is more than the intra-subject variability. Thus, the mean response time in multiple of TR was computed for each subject and the time segments for each of the 25 apps for a given subject were resampled to that mean response time value. By resampling in

such a fashion, we ensure time segments corresponding to all apps are of same length and the error due to stretching or compression of time series is minimized because of subject specific resampling. After resampling, all the time segments are concatenated together to obtain a modified representative time series for each ROI.

After the modified time series is obtained for each ROI, a Finite Impulse Response (FIR) analysis is performed to extract the hemodynamic response for each ROI. FIR modeling is a model-free approach to obtain the hemodynamic response from the time series data using the GLM framework [29]. The design matrix for FIR model consists of train of impulses at successive TRs. The total number of impulse repressors are equal to the duration of the stimulus in TRs. The weights for the FIR approach provide information about the shape of the hemodynamic response and in-turn the temporal response. Also, because the effect of hemodynamic delay is not modeled in the FIR model, the response for FIR model peaks few seconds after the actual impulse to account for the hemodynamic delay. The FIR modeling analysis was repeated independently for each subject to obtain the temporal response for all ROIs. As in case of dFC analysis, only the 14 subjects whose most ($\geq 13$ apps) response times were more than 10 seconds were considered for this analysis.

For group level analysis, the temporal response for different subjects were combined together. The temporal response for different subjects has different lengths. Thus, the temporal response for all subjects were again resampled to a fixed length of 100 time points. The second resampling normalized the temporal response on a scale of 0% to 100%, where 0% means the beginning of the stimulus and 100% means the end of stimulus. Once the temporal response for all the subjects were normalized, a sliding window analysis was performed to combine the response of different subjects into a single representation. To combine the response of different subjects, a time window of length 20% the entire app response duration is considered. Within that time window, peaks are identified in the temporal response of different ROIs. ROIs corresponding to the top 20 peaks, by amplitude of the peaks, are stored in bins. These stored ROIs are assumed to have the strongest BOLD response compared to other regions. The time window is then moved forward in time by 5% the app response duration and ROI bins are updated corresponding to the top 20 peaks in the time window. This process is repeated for all possible windows and all subjects.

Finally, the ROIs showing strong BOLD response, in a particular time window, for more than half of the subjects (bin value $\geq 8$) are considered to have a consistent temporal response in that time window. Other regions are discarded. Thus, thresholding the ROI bin values for all time windows give the combined spatiotemporal information. The combined spatiotemporal information can be represented in a 2D plot where one axis being the time and other being the ROI. The spatiotemporal information can be color-coded depending on the number of subjects showing simultaneous activation for a given region in a given time window (bin values). The extracted ROIs were then analyzed independently to identify their functionality and spatial location. The entire analysis was repeated two more times, one with only apps with response as YES and other with apps with response as NO. The idea here was to identify differences in ROIs for Yes and No responses.

## Results

### General linear model analysis

The spatial regions extracted for different contrasts in the GLM analysis are shown in Fig 3. The details about the clusters are in the appendix section of S1 File. The contrast for 'Yes > No' yields regions in the ventrolateral prefrontal cortex (VLPFC). The VLPFC is commonly involved in decision making tasks with higher reward [30]. On the other hand, the

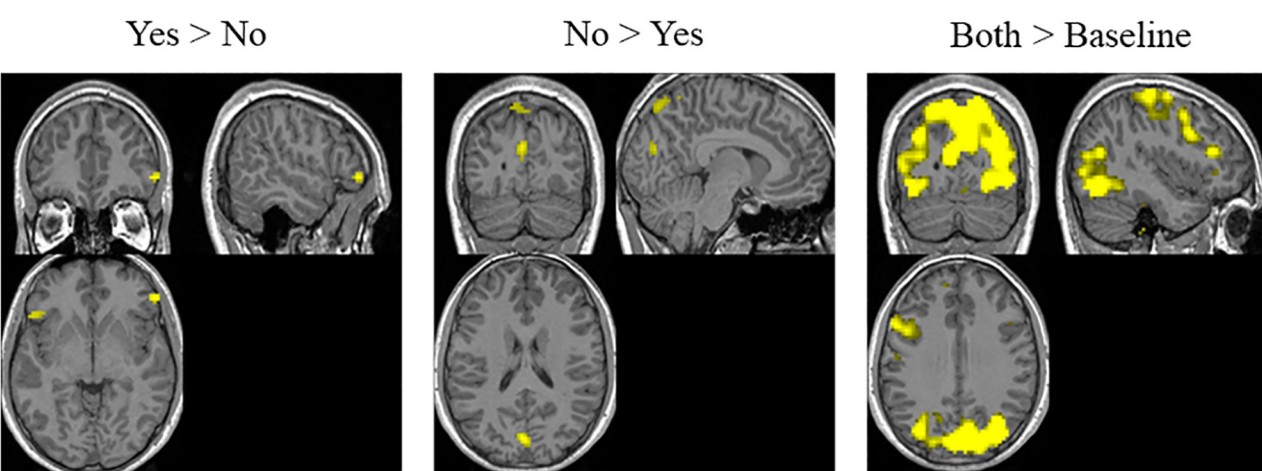

**Fig 3. Spatial regions corresponding to different contrasts in the GLM analysis.**

region for 'No > Yes' contrast lies in the cuneus and precuneus. The cuneus and precuneus activation have been shown previously to be overlapping in visual and memory search activities [31]. Precuneus is a part of the default mode network (DMN) which is involved when dealing with uncertainty in decision making [6]. Precuneus is also shown to be involved in response inhibition [32]. The regions of activation for the contrast 'Both>Baseline' are located in the left motor cortex, visual cortex and language processing regions, corresponding to the button press, the appearance of image of new app, and understanding description text respectively. To identify the neurodynamics of the entire decision-making process, the actual time when the decision was made in the brain should be known. Not only the decision-making is a continuous process [33] but also the exact temporal instance of decision-making cannot be measured using external devices. Thus, a functional connectivity analysis was performed.

## Functional connectivity analysis

The visualization of the group level dFC analysis for the beginning (first 10s) of the stimulus is shown in Fig 4. In the figure, the numbers on the outer ring of the circle represents IC numbers. The width of the line connecting different ICs signifies the number of subjects for which the high correlation was observed between the time series of two IC regions. Results are shown only for the pair of regions having significant correlation for more than half of the total subjects ($\geq$8). Finally, the color of the lines indicates positive (BLUE) or negative (RED) correlation between regions. The most consistent correlation is seen for IC14 (extra striate visual network) with IC34 (Dorsal attention network) and IC40. For both the connections, a strong positive correlation (14/34 r = 0.4516 ± 0.2856, 14/40 r = 0.3312 ± 0.2798) was observed for 13/14 subjects. IC18 (ventral visual network) and IC47 (dorsal visual network) both show a strong correlation (18/14 r = 0.386 ± 0.2912, 18/34 r = 0.5419 ± 0.2033, 18/40 r = 0.3312 ± 0.4465) with all the 3 networks mentioned above. Together IC2, IC14, IC18, and IC47 forms different parts of the visual cortices. A positive correlation is also observed between time series of the visual network and the primary motor network (IC1) (1/2 r = 0.441 ± 0.2503). The strong BOLD signal change in response to the button press followed by the change of visual stimulus causes a significant correlation between the time series of both motor and visual network.

A positive correlation is observed between many pairs of regions. First, posterior regions of the DMN which includes medial temporal regions (IC4) and the posterior cingulate gyrus

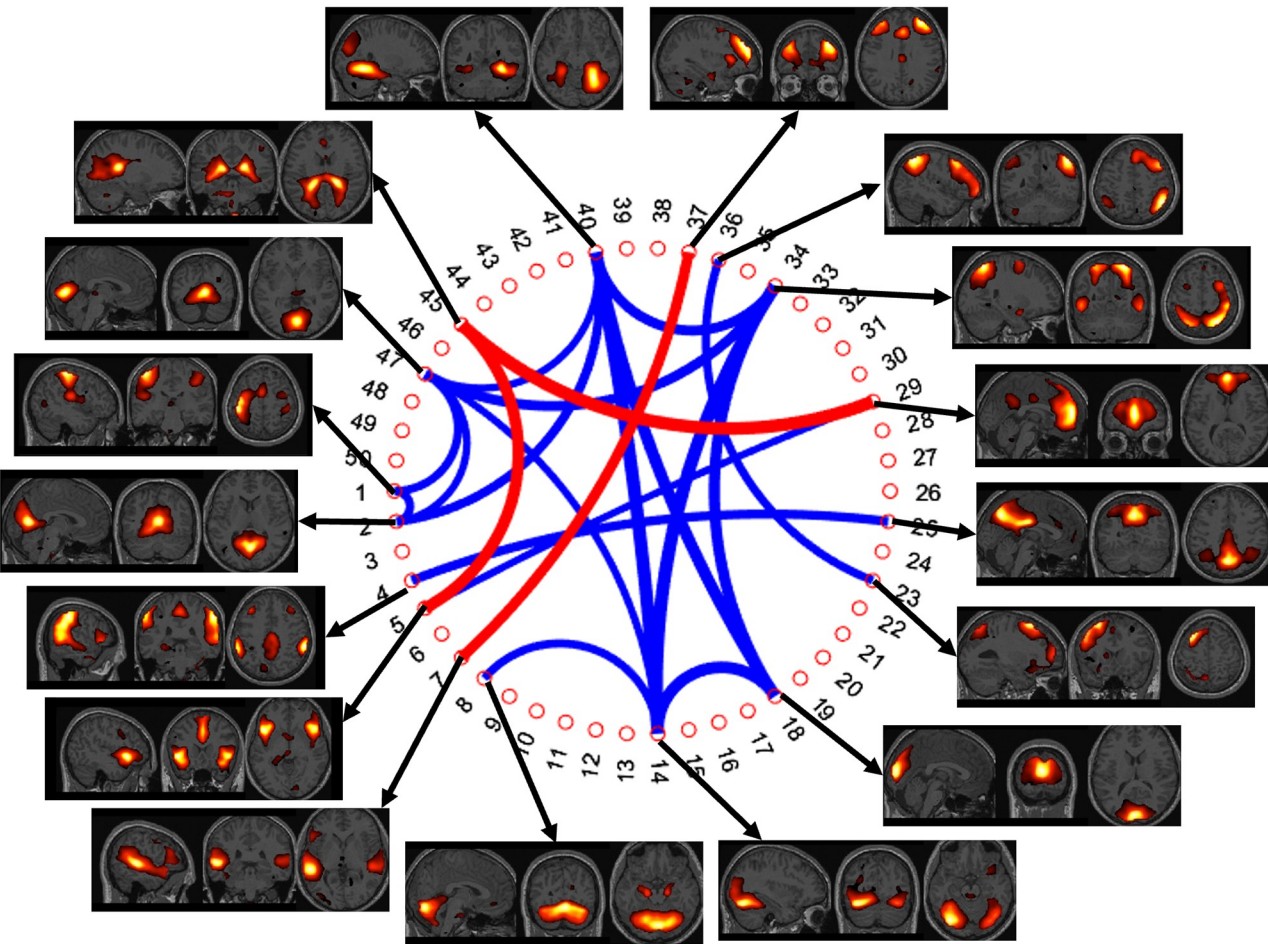

**Fig 4. Functional connectivity analysis result for first 10 seconds (beginning) of the app stimulus.** The number on the outer ring corresponds to IC numbers while the thickness of the line connecting pair of number shows the strength of correlation between the time series of the regions (blue = positive, red = negative). Thicker lines indicate correlation was observed for a greater number of subjects. The plot shown is thresholded at 8 subjects so the thinnest line is consistent correlation for 8 subjects and the thickest line is for consistent correlation in all 14 subjects.

(IC25) (4/25 r = 0.3713 ± 0.36). Second, left (IC23) and right (IC36) parts of the frontoparietal network (23/36 r = 0.4947 ± 0.2024). Along with that, the anterior parts of the DMN (IC29) shows correlation with the salience network (IC5) (5/29 r = 0.36 ± 0.352). The salience network comprises of the insula and Anterior Cingulate Cortex (ACC) [34]. Anticorrelation is observed between the executive control network (IC37) and IC7. Regions of IC7 are regions involved in language, semantic and visuospatial information processing. Anticorrelation is also observed between time series of IC45 with both the salience network and the anterior DMN. Spatially the IC45 regions doesn't correspond to any known brain networks.

The dFC analysis was also conducted for the end of the stimulus (last 10s). The connectivity map was similar to the once shown in Fig 4, except for few changes. Connectivity between some pairs of regions, which was not significant earlier, became significant and no connectivity was observed with some pairs where earlier there was significant correlation. Fig 5 shows the difference in functional connectivity for the beginning and the end of stimulus. The thin lines indicate connectivity only during the beginning of the stimulus while the thick ones indicate connectivity only towards the end of stimulus. It can be observed that most of the

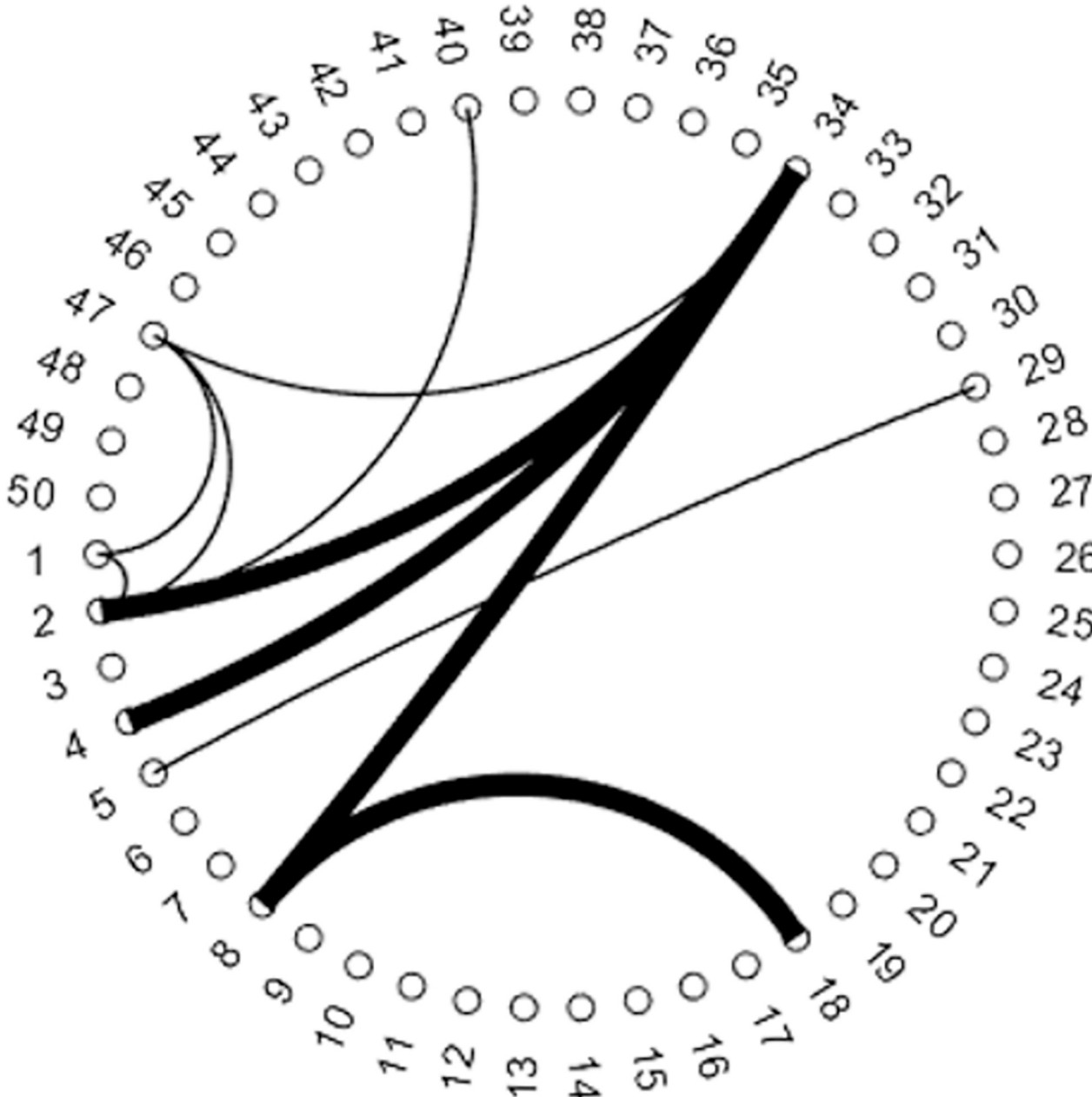

**Fig 5. Difference in the dynamic functional connectivity between the first 10 and last 10 second of the app stimulus.** Thin lines indicate connectivity only for the beginning of the stimulus while **thick** lines indicate connectivity only for the end of the stimulus. Functional connectivity between other pairs of regions is same as shown in Fig 4. (This figure is to be inferred along with Fig 4).

connectivity between the sensory input network (visual mainly) and the motor network (IC1, IC2, IC40 & IC47) is not present towards the end of the stimulus. The connectivity between the salience network and the anterior DMN is also not significant towards the end of stimulus. Significant connectivity is observed for the attention network with DMN and cerebellum.

The temporal response for different brain regions is shown in Fig 6. The most prominent response is seen the visual cortex region. The response in the visual cortex region is observed

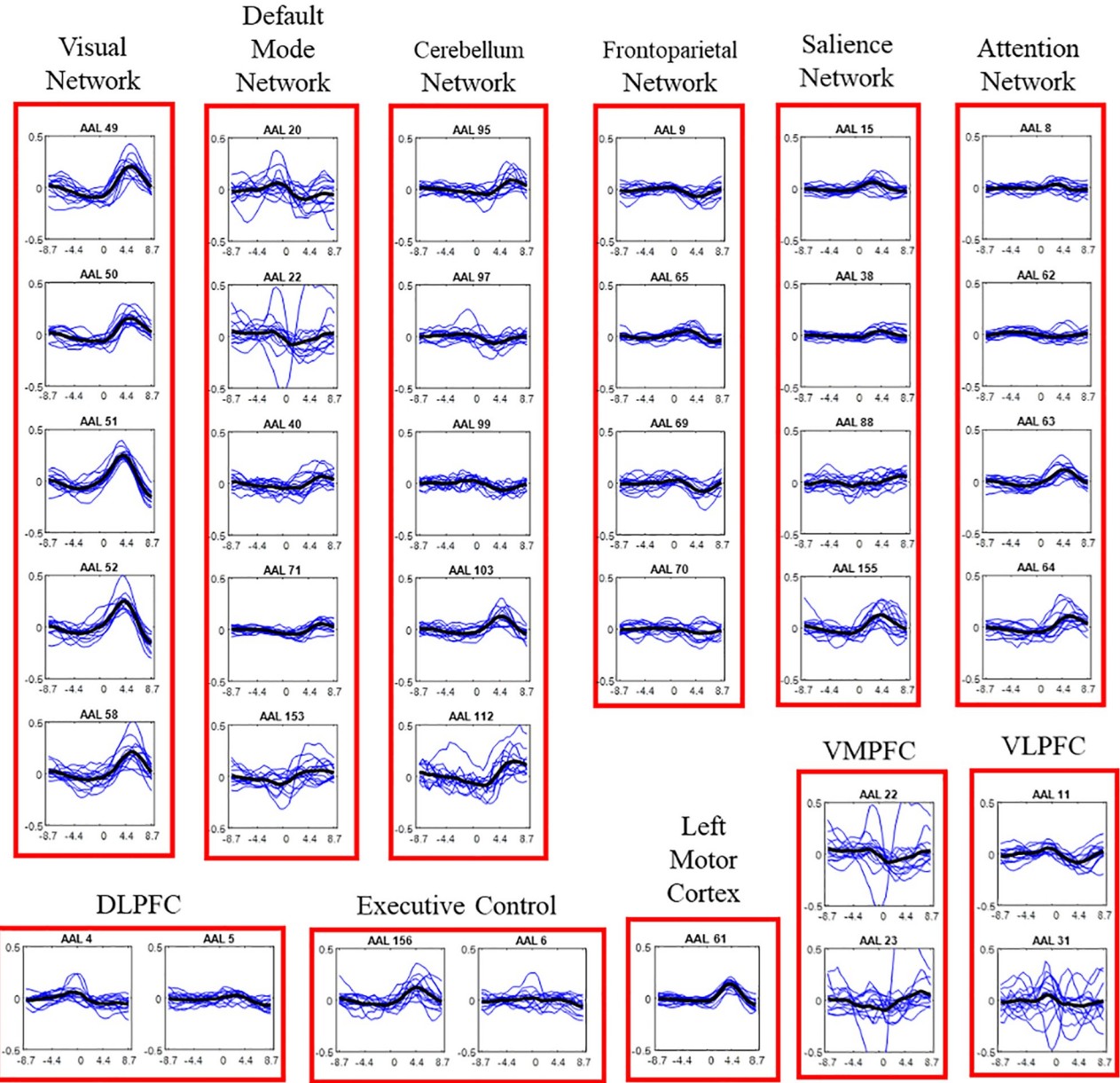

**Fig 6. Temporal response plots for ROIs in different functional networks.** The blue lines correspond to each subject's response while the black line is the mean over all subjects. The x-axis represents time centered at the instance of button press, with the entire duration ranging from 10 seconds before to 10 seconds after the button press. The y-axis shows the percent signal change w.r.t the mean intensity value for particular ROI.

after the beginning of a new stimulus while little to no response is observed towards the end of the stimulus. On the other hand, for the regions of DLPFC (dorsolateral prefrontal cortex), VLPFC and VMPFC (ventromedial prefrontal cortex), a positive response is observed before the end of stimulus which continues as a negative response after the beginning of the next stimulus. Some regions of the DMN and the cerebellum, shows a delayed response to the beginning of the stimulus. The variability in the temporal location of peak response and the duration of response is also observed within different regions of the visual cortex. For example, response for AAL 51–52 has a higher peak compared to other regions but the duration of the

response (width) is also smaller than the response in AAL 49, 50. Moreover, the temporal location of the peak response for AAL 58 is also a couple of seconds after the peak response in AAL 51–52. The response in the left motor cortex seems to peak around 4.4 seconds. Since the temporal responses are not corrected for hemodynamic delay, the temporal peak location is shifted by 4–5 seconds. Thus, the peak corresponding to the button press, indicating the end of trial, is observed during the beginning phase of the next trial. The difference in the response time of various regions suggest that the entire process can be broken down into various stages. The detailed inference on different stages of this process is shown in the results of the spatio-temporal analysis.

## Spatiotemporal analysis

The spatiotemporal activation information for all apps all subjects is shown in Fig 7. The vertical axis represents the AAL atlas ROIs while the horizontal axis represents normalized time. The time axis is represented in percentage. The beginning of the stimulus is denoted by 0% time while the end of the stimulus, instance of button press by the subject, is indicated by 100%. The different colors indicate the number of subjects showing consistent activation during a particular time window in a particular brain region. The plot shown here is thresholded to only show the regions which show consistent activation for at least half of the subjects ($>7$). The unthresholded plot (consisting of all ROIs) is shown in the appendix section of S1 File. Moreover, as FIR model was used for obtaining the hemodynamic response, the plot is also not corrected for the hemodynamic delay and shows the exact temporal location of the BOLD activation. In reality, a stimulus given to any brain region induces BOLD signal changes few seconds after the stimulus which is commonly known as the hemodynamic delay. A normal value for hemodynamic delay is anywhere between 4–6 seconds [35].

From the figure it can be observed that a large number of regions are involved at the beginning of the stimulus. The different spatial regions corresponding to different bands of the spatiotemporal plots are visualized in Fig 8. The most dominant activation is observed in AAL brain regions 48, 51 and 52. These regions corresponds to the primary visual cortex. The activation in these regions may be caused due to the presence of the visual stimulus. Slightly after that, in 20% - 50% time window, AAL regions 49, 50, 53, 54, 56, and 58 shows activation for more than 11 subjects. These regions correspond to secondary visual cortex V2 and the right visual cortex region V3. The main functions of region 58 involves visuospatial information processing and horizontal saccadic eye movements. Along with that, activation is also observed in the right fusiform region (AAL 60). Right fusiform is involved in integration of visual elements into perceptual wholes and visual memory task processing. The activation in this region may be because of the subjects reading the text corresponding to the apps, relating it to something similar seen before, and making sense of it.

The visual activation in most subjects lie within 10% to 40% time window. One of the reasons for obtaining a wider time window of activation may be due to different response times of subjects for different apps. The decision time for different apps range from 10 seconds (threshold) to 45 seconds. Corresponding to those decision times, 10% - 40% (10% of 45 = 4.5 and 40% of 10 = 4) gives an absolute time of about 4–5 seconds as the absolute response time. This response time is within the range for the hemodynamic delay observed in other fMRI studies.

Apart from the visual cortices, consistent response is also observed in the AAL region 61 between 10% - 30% time window. The region 61 corresponds to the left motor cortex. Given the fact that the decision responses were collected using button press with right hand, it is expected to observe consistent activation in the left motor cortex region. Moreover, little to no activation is observed in the right motor cortex. Between 10% to 40% of the decision time,

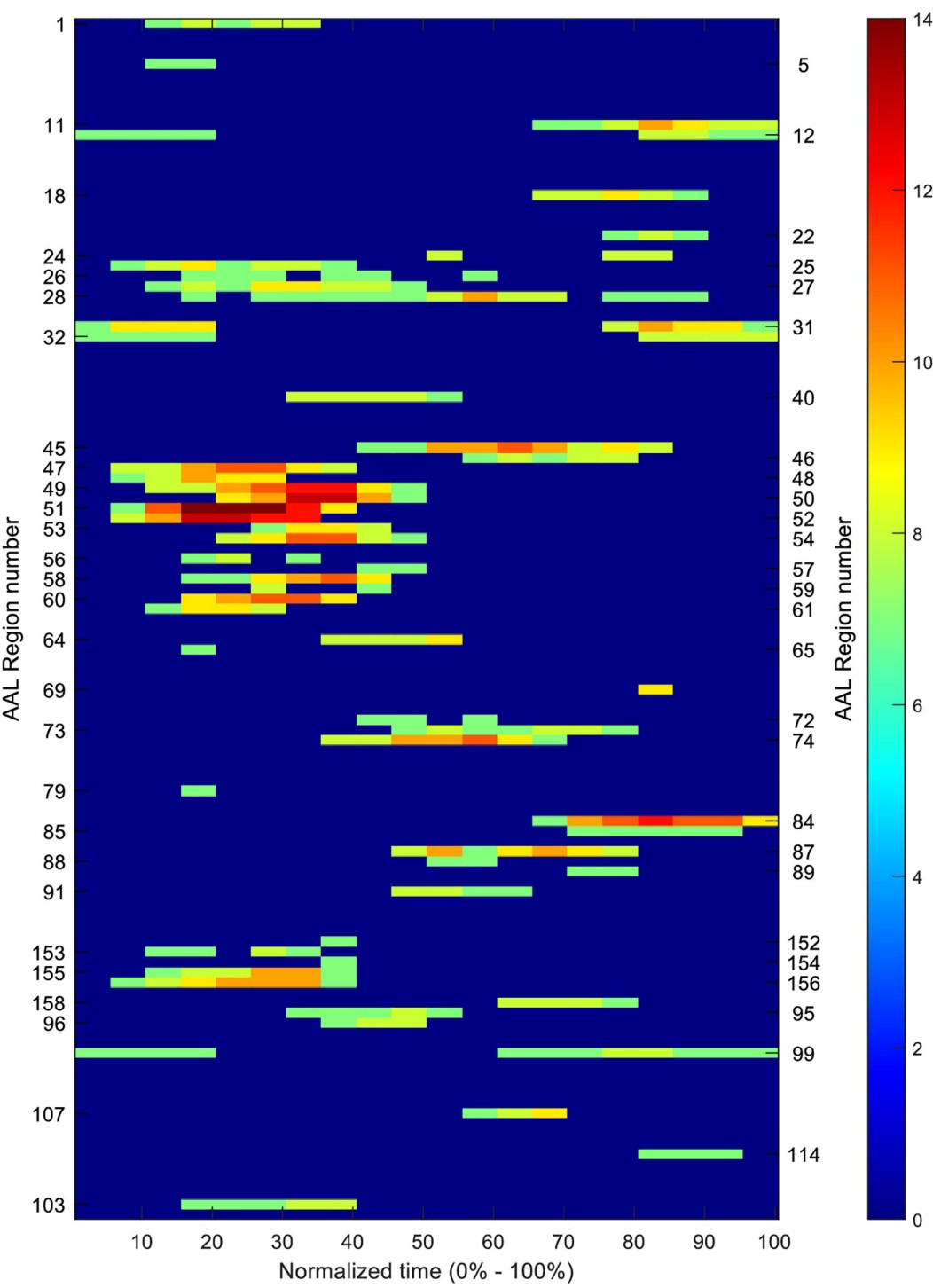

**Fig 7. The spatiotemporal activation plot.** The x-axis is normalized time with 0% indicating the start of stimulus and 100% indicating the end of stimulus. The y-axis shows AAL atlas ROI numbers. Each ROI number corresponds to a brain region. The color indicates the number of subjects for which significant activation is observed in each time window–ROI pair. The plot is thresholded at 7 so ROI activation consistent across at least 7 subjects is displayed.

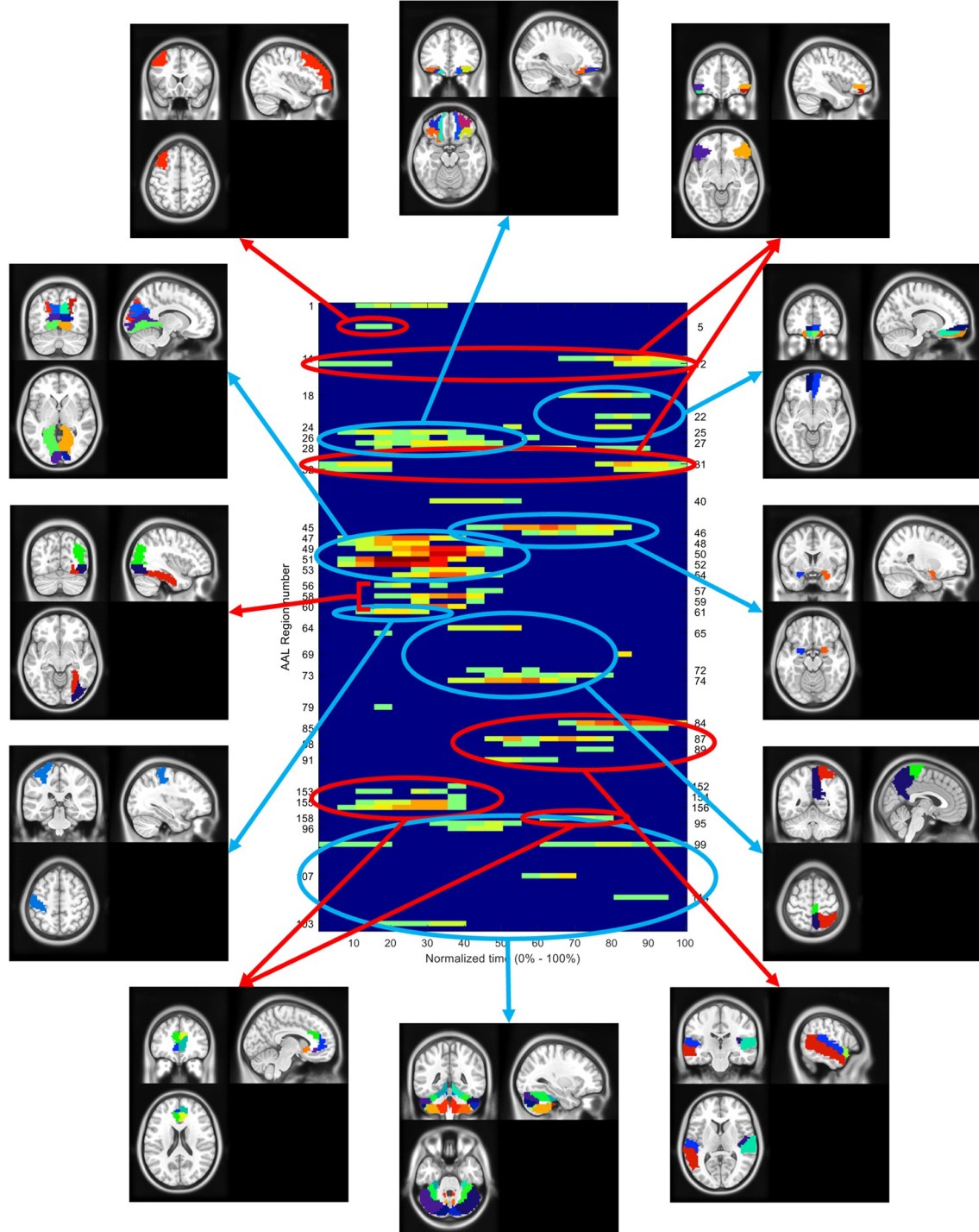

**Fig 8. Brain regions corresponding to significant ROIs in the spatiotemporal plots shown in Fig 6.**

activation is observed in the superior Anterior Cingulate Cortex (ACC) (AAL 155–156) and the Orbitofrontal Cortex (OFC) regions (AAL 25–28). The ACC region is involved in attention and conflict monitoring tasks [37–39] while the OFC regions are involved in reward network [52].

During the mid-temporal range, 35% - 70% of the time duration, activation is observed in amygdala (AAL 45–46), somatosensory association cortex (AAL 64), and premotor cortex (AAL 74). Previous studies have shown the involvement of these regions in visuospatial memory, recollection of past experiences, moral judgments, inferential reasoning, decision-making involving reward and inhibition. Activation is also observed in parts of the posterior DMN (AAL 40, 71–72). Before the end of the trials, activation was observed in left middle temporal gyrus (AAL 89), superior temporal gyrus (AAL 84–86), and prefrontal ACC (AAL 158). These regions have functional involvements in deductive reasoning, semantic processing, outcome evaluation and response inhibition. [53, 60]. The functionality of these regions shows their involvements in the process before a decision is made by the subject.

Towards the end of the trial, regions mainly in the frontal lobe including right dorsolateral prefrontal cortex (DLPFC) (AAL 5) and ventrolateral prefrontal cortex (VLPFC) (AAL 11–12, 31–32) got activated. These regions are known to be involved in the decision-making process [55] and possible responsible for the making the final decision of whether to install or not install the displayed app. One common characteristic about the activation of these region was that for some of the subjects, the peak activation was observed during the beginning of the trial (0% - 10%) as the time to peak gets carried to the next trial because of the hemodynamic delay. Carried over response is also observed for regions of the orbitofrontal cortex (OFC) (AAL 25–28).

The difference between the ROIs for YES and NO apps were also analyzed. (*YES and NO apps refers to the apps for which the response of the subject was yes and no* respectively.) Fig 9 shows the spatiotemporal activation for YES apps, NO apps and difference plot between YES and NO apps. The spatiotemporal activation plots corresponding to YES and NO apps shows much similarity with Fig 7. For the difference plot, only the regions showing difference in activation for more than half of the subjects are shown here ($|N_{YES}—N_{NO}| > = 7$). It is observed that may regions in the cerebellum (AAL 100, 107, 109,118) and the posterior DMN (AAL 40, 71, 72) show consistent activation for NO apps only. The regions in the DLPFC (AAL 3,5), VMPFC (19, 21, 22, 24), VLPFC (9, 11, 31, 32) and ACC (AAL 151) shows more activation only for YES apps towards the end of the stimulus.

The temporal response for the dominant regions showing difference for YES and NO apps were used to train a machine learning model to predict the subject's response for download intent. Correlation between time series of different regions was used as features to make the classification decision. The details about the machine learning technique is given in the appendix section of S1 File. The regions included were located in amygdala (AAL 69), ACC (AAL 151), left angular gyrus (AAL 69), posterior DMN (AAL71, 72), cerebellum (AAL 109, 117), DLPFC (AAL 5), VLPFC (AAL 11) and VMPFC (AAL 21). For a total of 10 ROIs, the total number of features were 45 (Pairwise correlation between 10 time series). Fivefold cross validation was performed on the model and the training and testing accuracies (mean ± std. dev.) were 67.3 ± 3.3 and 58.9 ± 4.9 respectively. The classification accuracies are not perfect, but it suggests that the difference in the brain response does predict the outcome of the app download decision to some extent. The involvement of the regions mentioned above in the entire decision-making process is described in detail in the next section.

## Discussion

The results shown above are an attempt to understand the neurodynamics of the technology adoption decision-making process for mobile application downloads. Decision making is a process that occurs over time involving various brain networks. By studying the temporal variation in the activation patterns in the brain, we attempt to build a better understanding of how

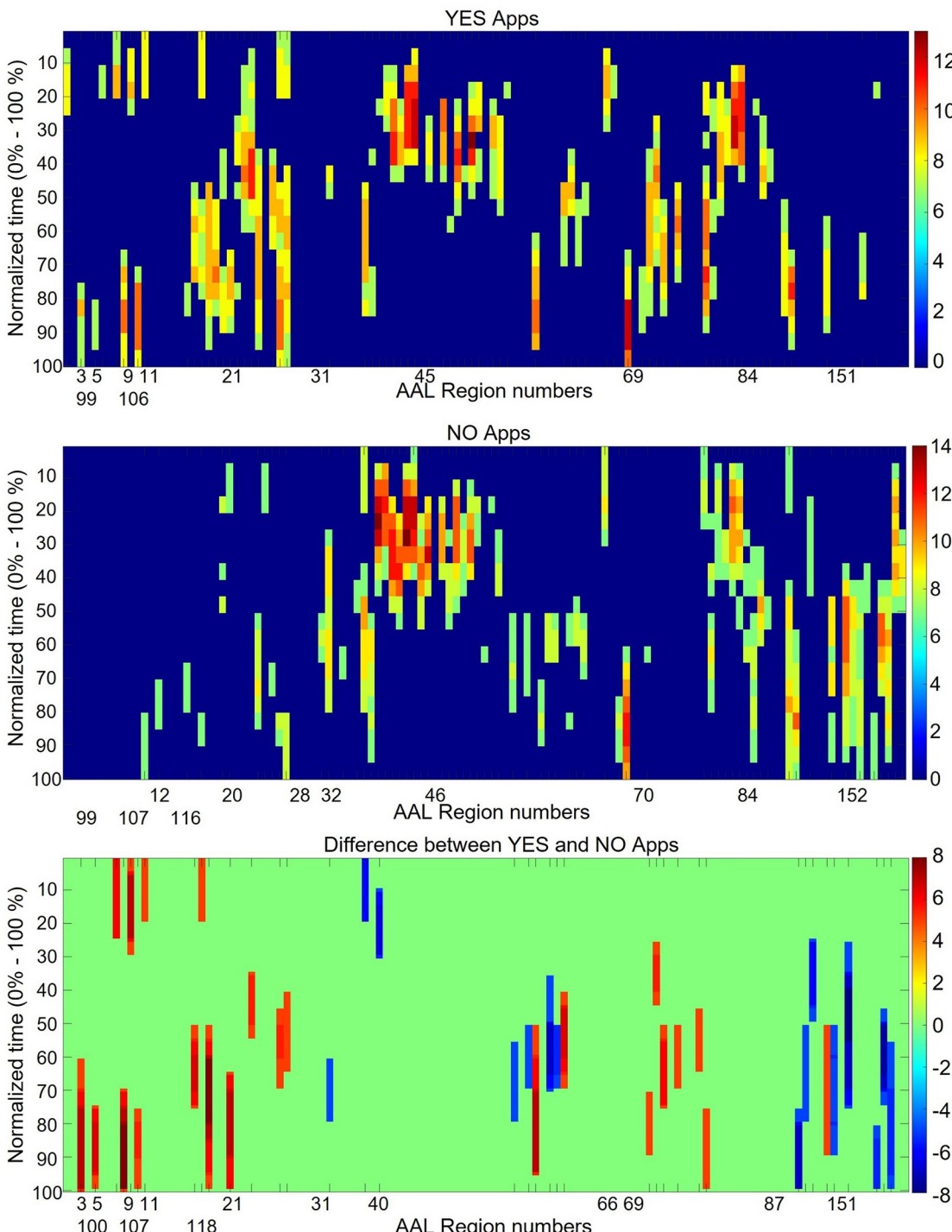

**Fig 9.** (top–middle) Spatiotemporal plots for only YES (top) or NO (middle) response to the apps. The axis is flipped w.r.t the plot in Fig 5 while keeping the same color coding. (bottom) The spatiotemporal difference plot indicating the difference between the ROIs which shows significant difference in response between YES and NO apps. The red lines indicate ROIs more active in YES apps while the blue lines indicate the ROIs more active in NO apps.

people make app downloading decisions. The entire neurodynamic process is summarized in Fig 10. The entire activation pattern can be broken down into various time segments and the functional information in different segments can be interpreted as follows.

The activation information for different regions in the spatiotemporal plot suggest a consistent process in the brain while deciding to download an app or not. The first step is to interpret the visual information presented in form of images and text which is observed as activation of the visual cortex regions. The activation observed in the visual cortex is one of the most common regions where fMRI activations are reported. In fact, the visual cortex activations are so consistent and strong that some early work on identification of the nature of the hemodynamic response function used visual cortex activations. The visual cortex activation at the beginning suggests the beginning of the information acquisition phase where the subjects look at the image and text of the new app and start to gather information about it. As described by Ploran et al., the process of decision making involves three distinct categories of brain regions depending on the nature of activation: sensory regions, evidence accumulator and recognition regions [33]. According to that, the visual cortex regions, primarily V1 and V2, falls under the sensory regions.

Immediately after the primary visual cortex activation, activation is observed in the region V3 of the visual cortex and the fusiform. The regions of V3 have been shown to be involved in horizontal eye movements for reading and visuospatial information processing. Again, it is intuitive that the activation in this region is because of the subjects reading the textual description for the apps. The right fusiform also shows involvement along with V3. Studies have shown that the fusiform, especially the right fusiform, shows higher level of activation when viewing similar objects seen before. The activation observed for the apps maybe due to the subjects relating the presented app with apps previously seen or interacted with.

During the same time, activation is observed in the superior ACC and OFC. Studies have shown the involvement of ACC in various functions during the decision-making process, two of the main ones being conflict monitoring and outcome evaluation [36]. Related to the app decision context, a likely reason for ACC activation is monitoring the conflict of whether to install the app or not [37–39]. The role of OFC has been shown to receive inputs from the sensory and somatosensory networks and generate a reward signal [11]. This reward signal is then forwarded to other areas like the amygdala, ACC and DLPFC. Thus, the ACC and OFC along with the right fusiform and V3 regions fall under the category of evidence accumulator regions.

Higher activation in the OFC may correspond to higher reward which may lead to activation in favor of the decision in VMPFC and DLPFC. The spatiotemporal activation for YES and NO apps clearly show a large difference in the OFC activation during the middle of the stimulus. A stronger response in the OFC, for YES apps, suggest a stronger reward signal which may be the starting point for a positive decision. The OFC response during the same time for NO apps is much weaker suggesting less reward signal and, maybe, uncertainty to download.

The results of the dFC analysis also shows a similar trend. During the earlier phase of the stimulus, high correlation is observed among the sensory input networks. Also, high correlation is observed between the sensory and the attention network. During the same time, the task negative network [40], i.e., the DMN, shows high correlation with the salience network. The salience network, the executive control network and the DMN together forms the triple network system [41]. The triple network theory suggests that the salience network mediates the activation and deactivation between the DMN and the executive control network [41, 42]. Insula, a part of the salience network, is responsible for integrating external information with internal emotional signal and initiate a switch between the DMN and the executive control

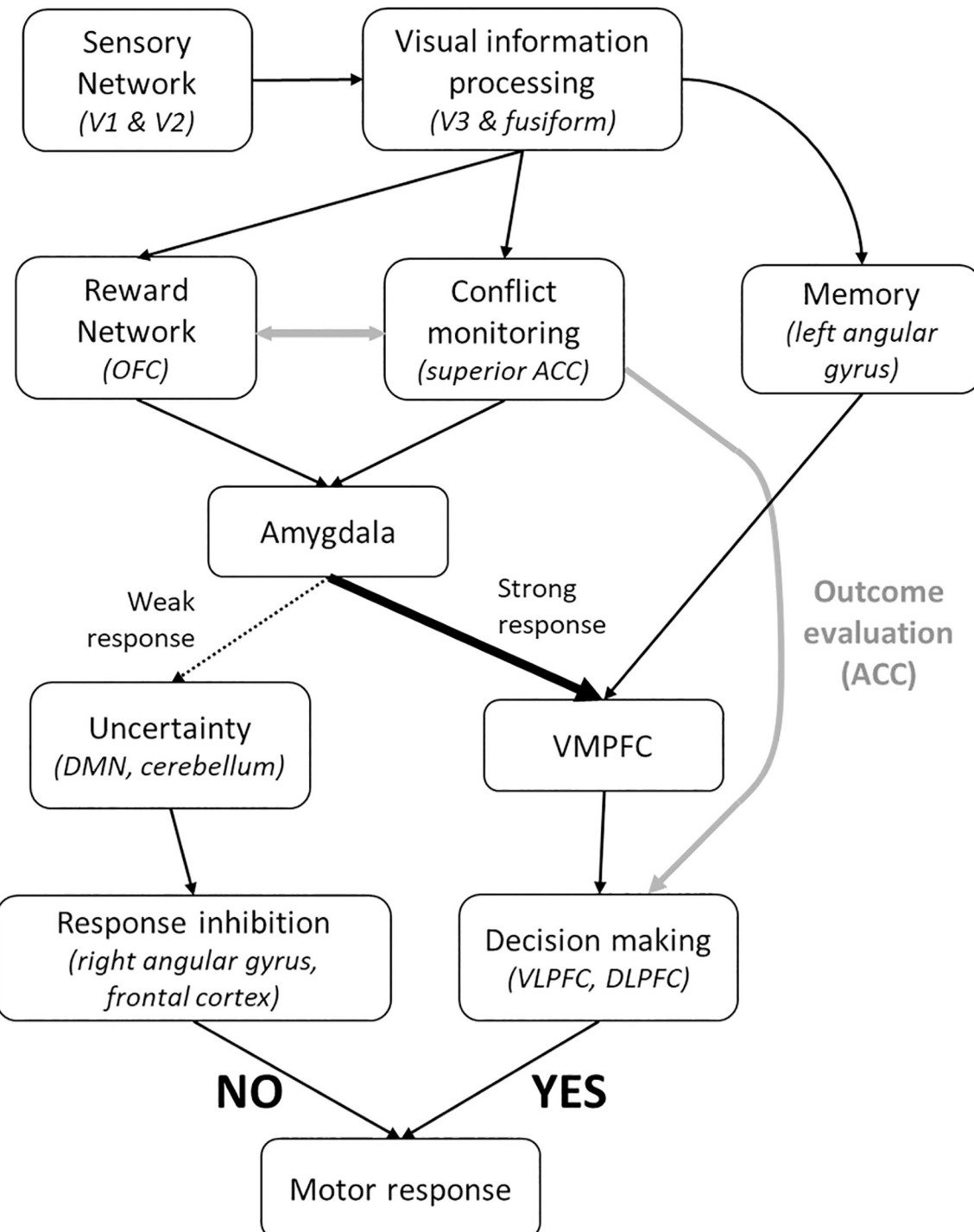

**Fig 10. Summary of neurodynamic response model for app download decision making.** Different boxes represent either different brain regions/ networks or region corresponding to different brain functions. The black arrows indicate communication between different brain regions. The direction of the arrow was inferred using concepts similar to Granger causality. Grey arrows indicate speculated communication between different brain regions. The dotted and the sold arrows going out of the Amygdala box indicate the relative strength of activation of the Amygdala (Dotted arrow–weak response and Solid thick arrow–strong response).

network [34, 43]. The difference between the dFC for beginning and the end of the stimulus shows the dissociation between the salience network and DMN suggesting the activation of DMN which is caused by the end of activation of the sensory network. The activation of the DMN is commonly associated with a deactivation of the executive control network, which is observed in the temporal response as a dip in the time series after a short peak. The correlation between the sensory input networks is also not observed in the dFC for the end of stimulus. The correlation of the attention network also reduces with sensory network and increases with the DMN and cerebellum network; both of which are shown to be involved in decision making related to uncertainty [6, 44].

During the middle of the stimulus, for the spatiotemporal response, the activation in the visual cortex subsides. The activation focus shifts to the dorsal DMN, amygdala, cerebellum and the temporal cortices. The DMN and cerebellum are shown to be involved in decision making under uncertainty conditions. DMN is also responsible for much of the automated decision-making process [6, 44]. Both the DMN and cerebellum activation is mostly observed in the apps for which the response was NO. The amygdala on the other hand is involved in emotions, value, and reward network [45, 46]. One important role of the amygdala is its link- age with the VMPFC. Amygdala activation is linked to initiate and influence the activation of VMPFC for decision making tasks [47]. The role of VMPFC is like a reflective system that reflects upon the inputs from the emotional and memory systems and combines them to initi- ate final decision. In absence of the input from the amygdala, the VMPFC response is not strong. The spatiotemporal plot for YES and NO app reveals that for YES apps, the amygdala response is stronger and thus a stronger response is seen in the VMPFC. Along with amygdala, strong response is also observed in the left angular gyrus for YES apps and not for NO apps. The left angular gyrus has been associated with episodic memory and episodic simulation [48– 50]. It may suggest that the higher response in the amygdala and OFC may relate to potentially higher reward from using the app and higher response in the left angular gyrus represents interaction/recollection with similar app in the past Together, higher response from the mem- ory and reward network combined in the VMPFC may lead to a positive (yes) decision to download the app. The positive response of the VMPFC may also be monitored by the ACC during the final phase of decision making that leads to much higher activation in ACC [39, 51, 52]. On the other hand, for lower amygdala response, the uncertainty network of the DMN and cerebellum activates, and the decision-making process is more uncertain and impulsive which leads to a negative (or possibly a random) response.

Towards the end of stimulus, the activation is observed in regions involved in the final deci- sion-making process; VMPFC, VLPFC and DLPFC [53–56]. In numerous literatures, the VMPFC has been associated with reward-based decision making while the DLPFC with effort based decision making [57–59]. The VMPFC and the VLPFC activations are more consistent and observed for a longer duration of time. Because of the hemodynamic delay, the BOLD response corresponding to the activation in the frontal cortex is observed after the beginning of the next stimulus. Along with the frontal cortex activation, strong activation is observed in the temporal cortex which is also reported to be involved in providing output dependent infor- mation for decision making [60, 61]. One important thing to observe from the activation towards the end of stimulus is the difference between right and left hemisphere activation for the frontoparietal network, especially the angular gyrus. Studies have shown the role of right angular gyrus in response inhibition [60, 62]. Comparing the activation pattern in the left (AAL 65, 67, 69) and right (AAL 66, 68, 70) angular gyrus for YES and NO apps, we observe that the right gyrus is more active for NO apps. Also, strong activation is observed only for NO apps in the right frontal cortex (AAL 12, 14, 32) which is also shown to be involved in response inhibition [55]. Strong activation in the right hemisphere for NO apps suggest that

the response inhibition is an additional response which drives towards not installing the app. Inhibition may serve as resolving the uncertainty (higher cerebellum and DMN activation) when the activation for the reward network (OFC) and the emotional network (amygdala) is low or absent, thus resulting in a negative response.

Finally, the response was indicated by a button press with the right-hand finger and thus a strong activation in the motor cortex is observed. Again because of the hemodynamic delay, the actual BOLD response is observed at the beginning of the next stimulus.

## Challenges and limitations

The fMRI experiment paradigm for the current study is neither a complete block design nor a complete event related design. Each instance of an app shows to the subject can be treated as a stimulus block but with different duration unlike the traditional fixed duration. On the other hand, onset of each of the different app can be treated as an event but the main difference here is that each event cannot be considered as impulses as the subjects first look at the image of the app, then read the description and then make any decision. Thus, the duration of each stimulus (apps) varies with subjects and apps. The main aim behind having such a complicated experiment paradigm was to observe the neurodynamics of natural decision making and technology adoption. A fixed time window for each app will force the subjects to make the decision within a given time frame. Different subjects take different time to make a decision, some may take a longer time while others can be quick. In having a fixed time window, subjects needing a longer decision time might rush to make a decision while the ones who make a decision early might wander off the task during the remaining time. In either case, the accuracy of the analysis reduces.

Another common practice in fMRI studies is to have a resting state period between instances of different stimulus. Resting state periods are often kept to serve as a baseline. One major change in our experiment paradigm was the omission of the resting state period between consecutive stimulus (apps). In many fMRI studies activations under two (or more) conditions is aggregated across time and then subtracted. However, it is known that the brain is not quiescent during the resting state. This is not a problem when subtracting aggregated data, but when looking at a time series of data the activation in the rest period will leak into the task period resulting activation patterns that are not actually part of the decision-making process.

The unconventional experiment paradigm has its own challenges. The major challenge was analysis using classical techniques like the generalized linear model. Because the experiment paradigm is neither a complete block design nor a total event related design, it is difficult to obtain a predictor time series. The beginning and ending of each stimulus can be used for obtaining a predictor time series but the regions responding to that would be dominated by the visual cortices, corresponding to the changing picture of the apps, or the motor cortex, _corresponding to the button press. Then when does the decision-making process occur? Is it 1 second before the button press? Or 2 seconds before the button press? It is almost impossible to predict beforehand the exact location of the decision-making process within the brain. Thus, an unorthodox approach was used for the analysis of the fMRI data. The results obtained were then verified using a machine learning deconvolution algorithm.

Another challenge lies in the temporal resolution of fMRI. On average the temporal resolution (sampling) of fMRI is a couple of seconds, which is a quite large compared to spontaneous processes taking place inside the brain. To improve the temporal resolution a bit, multiband pulse sequence was used for scanning which allowed for the sampling time to be as low as 0.5 seconds. Further reducing the sampling rate may in fact degrade the SNR and introduce

relatively higher frequency artifacts [63]. Moreover, the fMRI is an indirect measure of the neuronal activity. It measures the changes in blood oxygenation in response to neuron firing instead of neuronal activity. The convoluted BOLD response is comparatively slow and sluggish and can be captured using sampling rate possible by the available fMRI hardware.

The spatial resolution used for the analysis also has its own limitations. Atlas ROIs were used to indicate different regions of the brain. The representative time series were obtained by combining the time series for all the voxels inside any given region. It is possible that some regions might have inhomogeneity which might lead to averaging of different temporal responses into a single one. The ideal case would be to perform a voxel-wise analysis to extract only the very specific brain areas but with fMRI data having anywhere about 100,000 voxels, a voxel-wise analysis is not possible. A massive univariate analysis can be performed on a voxel-wise level but again it leads to the limitation of not knowing the exact timing of decision making (experiment paradigm) happen inside the brain. We, however, did tried to use sophisticated data manipulation techniques like ICA to identify functionally homogenous regions but a tradeoff still exists between functional homogeneity and the size of ROIs.

It should also be noted that we did not pay subjects for correct answers, because there were not verifiable correct answers. This may result in reduced engagement in the task. However, for the particular task of deciding whether or not to download an app, this make sense as around 95% of apps are free to download. Thus, while subjects may not have "skin in the game" in the experimental task, they also do not have "skin in the game" in the real world when they decide to download apps. While our task has ecological validity for this specific question, it is possible that for other decisions payment could change the activation patterns and change the neurodynamic path.

## Conclusion

This work developed a dynamic description of the brain networks involved in the app download decision. The dynamic description offers some advantages over a static model. A static model that averages neural activation over the entire time course of a decision increases the opportunity to detect activation that is ongoing. However, it does this at the expense of diluting the signal from processes that do not occur over the entire decision window. A dynamic model sacrifices power in detecting low level buy long duration activation in order to capture more of the short duration higher level activation. This generates a map of multiple activations over time, each of which may give rise to potential interventions that may enhance the decision-making process.

A dynamic model also shows the temporal ordering of brain processes. Some, but maybe not all, of the earlier processes provide inputs into the later processes. This can help generate conditional models of the decision-making process. That is to say that the processing in network B is in some way conditional on the earlier processing in network A. This is a well-known feature of many psychological decision-making models, but it is less well explored in neural models of decision making. Understanding which networks activate in which order can help to better understand the entire decision-making process.

While dynamic models allow researchers an alternative way to look at neurological responses, this work focuses on the app download decision in particular. This is one of the most important technological decisions that people make. Over one trillion apps have been downloaded and apps are the primary way most people interact with technology in the world. If one wants to get technology to the masses, especially poor and underserved people, an app is the way to do it. Thus, it is important to look at this decision specifically in order to facilitate uptake and dissemination of technology to the six and a half billion people with cell phones

worldwide. This work provides deep insight into the neural processes involved and many potential brain networks to consider for future interventions to help disseminate technology more effectively.

## Supporting information

**S1 File. This file contains all the supporting tables and figures.**
(DOCX)

## Author Contributions

**Conceptualization:** Fred Davis, Eric Walden.

**Data curation:** Harshit Parmar.

**Formal analysis:** Harshit Parmar.

**Investigation:** Eric Walden.

**Project administration:** Fred Davis, Eric Walden.

**Supervision:** Fred Davis.

**Writing – original draft:** Harshit Parmar, Eric Walden.

**Writing – review & editing:** Fred Davis, Eric Walden.

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
