## [Decision Letter · Decision Letter 0]

19 Aug 2022

PONE-D-21-41018Understanding the Neurodynamic Process of Decision-Making for Mobile Application Downloading.PLOS ONE

Dear Dr. Walden,

Thank you for submitting your manuscript to PLOS ONE; I sincerely apologise for the unusually delayed review timeframe. Your manuscript has been assessed by one reviewer, whose comments are appended below with additional feedback in the attached document. After careful consideration of the reviewer's comments, we feel that your study has merit but does not fully meet PLOS ONE’s publication criteria as it currently stands. Therefore, we invite you to submit a revised version of the manuscript that addresses the points raised during the review process. In addition, per PLOS policy, please ensure that all supportive data are provided as part of the revised manuscript, and that the machine learning method is shared in an accessible manner (for more information, please see https://journals.plos.org/plosone/s/materials-software-and-code-sharing#loc-sharing-code). Please note that we have only been able to secure a single reviewer to assess your manuscript. We are issuing a decision on your manuscript at this point to prevent further delays in the evaluation of your manuscript. Please be aware that the editor who handles your revised manuscript might find it necessary to invite additional reviewers to assess this work once the revised manuscript is submitted. However, we will aim to proceed on the basis of this single review if possible.

Again, my sincere apologies for the delay. We look forward to receiving your revised manuscript.

Kind regards,

Emily Chenette

Editor in Chief

PLOS ONE

Journal Requirements:

7. Please upload a new copy of Figures 1 and 9 as the detail is not clear. Please follow the link for more information: https://blogs.plos.org/plos/2019/06/looking-good-tips-for-creating-your-plos-figures-graphics/" https://blogs.plos.org/plos/2019/06/looking-good-tips-for-creating-your-plos-figures-graphics/

8. We note that Figure 1 in your submission contain copyrighted images. All PLOS content is published under the Creative Commons Attribution License (CC BY 4.0), which means that the manuscript, images, and Supporting Information files will be freely available online, and any third party is permitted to access, download, copy, distribute, and use these materials in any way, even commercially, with proper attribution. For more information, see our copyright guidelines: http://journals.plos.org/plosone/s/licenses-and-copyright.

Reviewers' comments:

Reviewer's Responses to Questions

**Comments to the Author**

1. Is the manuscript technically sound, and do the data support the conclusions?

Reviewer #1: Partly

2. Has the statistical analysis been performed appropriately and rigorously? 

Reviewer #1: Yes

3. Have the authors made all data underlying the findings in their manuscript fully available?

Reviewer #1: No

4. Is the manuscript presented in an intelligible fashion and written in standard English?

Reviewer #1: Yes

5. Review Comments to the Author

Reviewer #1: This paper is about an exploration of the decision making process during fictive mobile app downloading, as a proxy for technology acceptance/adoption (TA). The aim is to suggest a neural model for the different consequtive phases in the decision making (DM) process. The paper is clearly written, carefully explaining the methodology and results. There is also a section with “challenges and limitations”, where the authors, in an insightful way, describe what they perceive are difficulties with their study, and where more research is called for.

My main concern with this paper is that the fictive app download is quite artificial, and may say little about the decision making process in the brain, considering a realistic situation when choosing apps to download. The decision to download an app or not may be considered arbitrary in some paradigms, since there is no experimental reward (i.e. skin in the game) provided as in e.g. (Maoz et al., 2019). This may weaken possible emotional or reward responses. While reflecting decision-making on app downloading, generally disinterest in the app (decision to act) or daydreaming may be more correlated with ‘No’ and may also drive a less salient decision. The “default mode” towards a ‘Yes’ could just reflect an inclination to choose more or less all apps, which are not perceived as bad or totally useless. If a cost of some sort (e.g. a limited budget or storage space).

It is also questionable if the fictive app download could actually be considered a TA, as there is no testing of the apps, and it is not clear what the Yes or No implies, if the subjects have understood the apps, and think they may be useful, or whether they just base their “decisions” on look, on mood, or plainly randomly. The reasons for answering Yes or No may be vague and unclear, also given the compressed description of the apps during the experiment.

A major problem is that all the data and other information is not (yet?) available, and the Appendix referred to in the text seems to be missing.

While the results and conclusions made are not totally convincing, the paper is a good attempt to trace the different steps in the decision making process of the brain, and thus opens up for further studies of this sort.

6. PLOS authors have the option to publish the peer review history of their article (what does this mean?). If published, this will include your full peer review and any attached files.

Reviewer #1: **Yes: **Hans Liljenström

---

## [Author Response · Author response to Decision Letter 0]

6 Nov 2022

Rebuttal Letter

Editorial points

We have moved this statement from “Subjects” to “Method”: “Written consent was obtained from all the participants prior to commencing the experiment. All the participants were also screened for MRI safety with a separate MRI safety screening form. This study was approved by the Texas Tech Human Research Protection Program with number IRB2016-557. ”

We did not receive any grant funding. This was paid for out of a faculty professorship. 

The entire dataset has been uploaded to ‘Texas Data Repository’. The link to the dataset is as follows: https://doi.org/10.18738/T8/QICUQB

The link to the dataset is also provided in the manuscript. 

We had this in the subjects section and have moved it to the methods section.

7. Please upload a new copy of Figures 1 and 9 as the detail is not clear. Please follow the link for more information: https://blogs.plos.org/plos/2019/06/looking-good-tips-for-creating-your-plos-figures-graphics/" https://blogs.plos.org/plos/2019/06/looking-good-tips-for-creating-your-plos-figures-graphics/

A high-resolution TIF images have been uploaded. The figures in the manuscript have also been changed to make it high resolution.

8. We note that Figure 1 in your submission contain copyrighted images. All PLOS content is published under the Creative Commons Attribution License (CC BY 4.0), which means that the manuscript, images, and Supporting Information files will be freely available online, and any third party is permitted to access, download, copy, distribute, and use these materials in any way, even commercially, with proper attribution. For more information, see our copyright guidelines: http://journals.plos.org/plosone/s/licenses-and-copyright.

The figure contained screenshot of an actual app that was used as a stimulus. The screenshot of the apps has been blurred so that it is no longer identifiable.

Reviewer Comments

General comments

Reviewer #1: This paper is about an exploration of the decision making process during fictive mobile app downloading, as a proxy for technology acceptance/adoption (TA). The aim is to suggest a neural model for the different consequtive phases in the decision making (DM) process. The paper is clearly written, carefully explaining the methodology and results. There is also a section with “challenges and limitations”, where the authors, in an insightful way, describe what they perceive are difficulties with their study, and where more research is called for.

Thank you for the kind and positive comments.

My main concern with this paper is that the fictive app download is quite artificial, and may say little about the decision making process in the brain, considering a realistic situation when choosing apps to download. The decision to download an app or not may be considered arbitrary in some paradigms, since there is no experimental reward (i.e. skin in the game) provided as in e.g. (Maoz et al., 2019). This may weaken possible emotional or reward responses. While reflecting decision-making on app downloading, generally disinterest in the app (decision to act) or daydreaming may be more correlated with ‘No’ and may also drive a less salient decision. The “default mode” towards a ‘Yes’ could just reflect an inclination to choose more or less all apps, which are not perceived as bad or totally useless. If a cost of some sort (e.g. a limited budget or storage space).

The point about skin in the game is well taken. We have added a discussion of these issues to the paper. We were very impressed with Maoz et al paying $1000. Certainly, there may be differences in small money decisions and large money decisions. However, in the specific case of downloading apps, there is great ecological validity because most apps are, in fact, free. Around 96% of Google Play and 94% of App store apps are free as of Jul 2022 https://www.statista.com/statistics/263797/number-of-applications-for-mobile-phones/. We agree that this may weaken emotional reward and everything that you suggest. But this is the real decision that people make. It is a small, low cost, decision. But in aggregate it is a hugely important decision. On average, the average citizen of Earth decides to download an app every two weeks (Blair, 2021). They also decide not to download an unknown number of apps. Taken together the app download decision is the most frequent information technology (IT) adoption decision (Blair, 2021). Moreover, in terms of revenue apps are more important than traditional software (Stancheva. 2022). 

So we do agree with everything you say, but this is really a decision that people make in the world a lot and these are the actual stakes they play for most of the time. So, the decision has a great deal of ecological validity. It mirrors the real world, and the issues you raise are the same issues that face real decision makers in the real world. 

Here is what we have added:

“It should also be noted that we did not pay subjects for correct answers, because there were not verifiable correct answers. This may result in reduced engagement in the task. However, for the particular task of deciding whether or not to download an app, this make sense as around 95% of apps are free to download. Thus, while subjects may not have “skin in the game” in the experimental task, they also do not have “skin in the game” in the real world when they decide to download apps. While our task has ecological validity for this specific question, it is possible that for other decisions payment could change the activation patterns and change the neurodynamic path. ”

It is also questionable if the fictive app download could actually be considered a TA, as there is no testing of the apps, and it is not clear what the Yes or No implies, if the subjects have understood the apps, and think they may be useful, or whether they just base their “decisions” on look, on mood, or plainly randomly. The reasons for answering Yes or No may be vague and unclear, also given the compressed description of the apps during the experiment.

Again, we agree with you, and have added a discussion of this to the paper. People randomly download apps on a whim all the time. Some, they never use, some they delete, and some they keep. But they should be good at looking at an app and making that decision. Moreover, the actual stimuli that we use are based on the actual information that people see in the real world while making these decisions. This is what the app makers choose to share. These are real apps. The descriptions are compressed due to the limitations of the scanner, but realistically people likely do not read all the words that an app description provides. 

A major problem is that all the data and other information is not (yet?) available, and the Appendix referred to in the text seems to be missing.

We are sorry about this oversite; our understanding is that we were to do this after the paper was accepted and before it was published.

The dataset is uploaded to https://doi.org/10.18738/T8/QICUQB

While the results and conclusions made are not totally convincing, the paper is a good attempt to trace the different steps in the decision making process of the brain, and thus opens up for further studies of this sort.

Thank you, our goal was to trace the different steps in this specific decision-making process and to create a road map for more studies of this sort. 

 

Specific comments and questions

When the authors talk about neurodynamics, it should be noted that it is macroscopic neurodynamics, and not the mesoscopic neurodynamics perhaps mostly used with EEG in cognitive tasks. As the authors recognize, fMRI has a rather low temporal resolution in the order of seconds, which cannot capture the fast dynamics in many decision making cases (c.f. the Libet type of experiments on volition). Hence, it is difficult to trace causal pathways involving neural events or processes at faster time scales.

I agree. We know fMRI is low temp hence we used multiband (with a temporal acceleration of 6) to obtain sampling time of 0.5 s. It is still not as good as millisecond resolution but the observed BOLD response itself is limited by hemodynamic convolution and thus very fast sampling (millisecond) may not provide significant improvement. Moreover, having a fast sampling rate may introduce more artifacts and reduce the overall SNR of the observed BOLD signal.

Some of the limitations were already discussed in the ‘Challenges and Limitations’ section and it has also been modified to add a few more lines and one more citation. 

It is clear and expected to see initial activation in sensory (visual) and attention areas, but it is less clear where and when any decision is taken, even though PFC areas are involved. 

Dynamics were observed in the cerebellum and amygdala but apparently not in the thalamus or basal ganglia. How about the SMA or pre-SMA, which have been reported to play an important role in decision making and volition? Why are these areas not included or displayed in the study?

We believe that the process of decision making is not sudden but occurs over time involving various regions of the brain. Thus, we did not pinpoint to any single time instance or brain region as the decision-making moment. Our goal was to trace the different steps in this specific decision-making process and to create a road map for more studies of this sort. 

The omission of resting states between choices is a bit problematic, as also the authors recognize. It seems quite clear that there will be some overlap/leakage between the activity of previous and the following app presentation/decision, and this may distort the activation patterns considerably. Perhaps the advantage with resting state periods exceeds the disadvantages? It could be worth comparing both methods. 

We agree to the fact that omission of the resting state period is debatable, and hence we have added a comprehensive discussion of that in the ‘Challenges and Limitations’ section. We are aware of the fact that some of the activation from the previous trial will leak into the next trian and it has been clearly stated in the paper. 

However, there are two main reasons for not having the resting state period between trials. First, the primary focus of the analysis was not on obtaining any contrasts but to identify the causality in the decision-making process. Resting state periods are typically good when comparing various stationary conditions. Second, we wanted to keep the actual trial as close to the real app scrolling experience as possible. In real world, there is no rest period when looking for apps on the app store. 

There seems to be a rather low number of (eligible) subjects for the experiment, making any statistical analysis problematic, and conclusions questionable. It is also not stated whether all those subjects were right-handed, which may have an effect on the activity in motor cortices. 

The whole idea behind this study was to try and map out various steps involved in the decision-making process using rather unorthodox experiment design and analysis techniques. It was an exploratory study and thus the low number of subjects. Initially we anticipated to have all 20 but unfortunately had to discard some of those as we feared that data from those participants may introduce artifacts. 

Irrespective of their dexterity, all the participants responded using the index and middle fingers of the right hand and thus strong activation was only observed in the left motor cortex and not in the right. 

The types of apps are not reported, neither there attractiveness for various purposes, nor the (fictive or real) prices, which all may influence the choices by the subjects. The familiarity or previous experience with the same or similar apps may drastically affect the decision and decision time. It would have been good if the authors had displayed the apps used in the experiments, and preferably also the subjects preferences/history with regard to similar apps. 

I agree that same/similar apps may create a bias in the decision. Unfortunately, no post scan information was collected from the participants regarding previous app exposure. However, a modified study was conducted later where we did ask the participants post scan about their exposure to the apps and on an average participant had only used 3 out of the 50 apps (max 8 out of 50). Even though, post scan app exposure was not measured for this study, we assume it would be similar to the study where we measured it. 

We would like to add the details of all the apps in the appendix section but cannot put any screenshot dur to copyright issue. We also had to modify figure 1 and blur the app screenshots due to PLOS copyright criteria. 

Data and other material is referred to an Appendix, which appears to be missing. 

Appendix added. Link to data also added.

GLM, as other acronyms or abreviations, should be spelled out the first time it is mentioned. 

Most of the acronyms were spelled out upon their first occurrence. A few were left which have been added. 

The machine learning method used for prediction is not explained or described, which would have been desired, perhaps in an appendix.

The details about the machine learning technique have been added to the appendix section.

The conclusions are a bit vague and not totally based on the presented results. That the largest activation is in the visual cortex is not surprising and is likely not significant for the decision making process. Instead, it is probably just reflecting the reading/perception of the app description. It could have been good to compare the activity in a situation without a decision following the visual presentation of the app. The authors also claim the decision can be predicted toward the middle of the process, but that is not really demonstrated. 

We have rewritten the conclusion. 

The point about comparing viewing with and without a decision is a very astute observation and it highlights the value of doing a dynamic model in this case. Consider if we showed a face and then asked if it was attractive. Then we showed a face and did not ask anything. It is very likely that the subject would automatically evaluate the attractiveness of the face even if we did not ask them to do it. You could instruct a person to NOT evaluate the attractiveness of the face, but realistically someone could probably not suppress that reaction. 

Comparing the download decision to just looking at the images with no download decision was our first inclination. The problem is that we do not know what someone does when they look at an app description without later being asked if they would download it. It seems that the default action would be to evaluate it and consider downloading it. We could give an instruction to NOT think about downloading it, but they may also not be able to suppress the evaluation of the downloadability of the image. 

Then we thought about giving them a distractor task. For example, look at this app description and make sure it is formatted correctly. There are two problems here. First, it is very hard to think of a task (other than evaluating the downloadability of the app) that makes sense when seeing a description of an app. This suggests that evaluating downloadability might be the default. Second, it is not clear what the BOLD contrast would even mean. If we compare the question of “would you download this” to the question of “is it formatted correctly”, then what would an activation in area X mean. Literally it means that there is higher activation for condition 1 than condition 2, but it does not mean that area X is relevant to the decision to download an app. It could be that there is no reaction for downloading in area X, but a deactivation for formatting. Similarly, if there is no activation in area X then does that mean that it is not important for downloading or that it is just equally important for downloading and formatting.

Fundamentally we need a condition where we say look at this and do all the stuff that you would normally do when looking at an app description, except deciding if you would download it. But if we give another task (like formatting) then we are asking them to some other task and maybe that other task also requires some of the same processing as the app download task. 

For this particular task and probably for many other tasks, there is no clear contrast that would make sense for BOLD analysis.

Figures and tables

The figures are in general good and descriptive. However, some of the figures are confusing or hard to understand:

Fig. 2 has an insufficient legend. 

The caption for figure 2 has been modified to make the interpretation of the figure easier.

In Fig. 3, Yes>No might represent activation of language centers in VLPFC, possibility indicating a more careful read of the description. But, this may not represent the actual decision making. One of two activated regions for No>Yes wasn't mentioned. It looks like visual cortex, perhaps V3 or MT but between hemispheres. Is this an artifact?

The second activation region belongs to the Cuneus. The details about the activation clusters are given in the appendix section while a few lines and one reference was added to talk about the second activation region. 

In Fig. 6 it is difficult to see or follow temporal/causal pathways, and in particular where the decision is taking place. It also seems that the (left) motor cortex peaks about 4.4 secs after the button press, which may be an effect of the BOLD delay, but should be explained. 

There is no causal pathway described in figure 6. It just shows the temporal response of different brain regions/networks. It is possible for a single brain network to consist of multiple AAL ROIs, which are grouped together in a single red box. 

The hemodynamic delay is discussed briefly in the next section (spatiotemporal analysis) but a couple of lines have been added here to explain the delay in the temporal peak for left motor cortex response. 

In Fig. 10, the different boxes represent different brain regions, according to the legend, but in fact there is a confusing mixture of brain regions and functions in the figure. 

The caption to figure 10 has been modified to include both brain regions and functions.

A reference table with the used AAL codes and brain areas might help interpretation.

A reference table containing names of all 170 AAL brain region has been added to the appendix section. 

References

The authors are suggested to consider any or all of the following articles for further references:

1. Frank, M.J. Claus, E.D. (2006) Anatomy of a decision: striato-orbitofrontal interactions in reinforcement learning, decision making, and reversal. Psychological Review. 113(2):300-26.

2. Hassannejad Nazir, A. & Liljenström, H. (2015). A cortical network model of cognitive and emotional influences in human decision making. Biosystems 136:128-141. doi.org/10.1016/j.biosystems.2015.07.004

3. Levine, D.S (2012) Neural dynamics of affect, gist, probability, and choice. Cognitive Systems Research. Volume 15-16 : 57-72.

4. Maoz U, Yaffe G, Koch C, Mudrik L (2019) Neural precursors of deliberate and arbitrary decisions in the study of voluntary action. Elife 8:e39787. https:// doi. org/ 10. 7554/ eLife. 39787. 001

---

## [Decision Letter · Decision Letter 1]

23 Nov 2022

Understanding the Neurodynamic Process of Decision-Making for Mobile Application Downloading.

PONE-D-21-41018R1

Dear Dr. Walden,

We’re pleased to inform you that your manuscript has been judged scientifically suitable for publication and will be formally accepted for publication once it meets all outstanding technical requirements.

Kind regards,

Wang Zhan, Ph.D.

Academic Editor

PLOS ONE

Additional Editor Comments (optional):

Reviewers' comments:

Reviewer's Responses to Questions

**Comments to the Author**

1. If the authors have adequately addressed your comments raised in a previous round of review and you feel that this manuscript is now acceptable for publication, you may indicate that here to bypass the “Comments to the Author” section, enter your conflict of interest statement in the “Confidential to Editor” section, and submit your "Accept" recommendation.

Reviewer #1: All comments have been addressed

2. Is the manuscript technically sound, and do the data support the conclusions?

Reviewer #1: Yes

3. Has the statistical analysis been performed appropriately and rigorously? 

Reviewer #1: I Don't Know

4. Have the authors made all data underlying the findings in their manuscript fully available?

Reviewer #1: Yes

5. Is the manuscript presented in an intelligible fashion and written in standard English?

Reviewer #1: (No Response)

6. Review Comments to the Author

Reviewer #1: It appears the authors have considered all comments, although the missing Appendix is still not attached to the final version. Given it contains all the information stated in the authors' letter, I believe the paper is now acceptable for publication.

7. PLOS authors have the option to publish the peer review history of their article (what does this mean?). If published, this will include your full peer review and any attached files.

Reviewer #1: **Yes: **Hans Liljenström

---

## [Editor Report · Acceptance letter]

28 Nov 2022

PONE-D-21-41018R1 

Understanding the Neurodynamic Process of Decision-Making for Mobile Application Downloading. 

Dear Dr. Walden:

I'm pleased to inform you that your manuscript has been deemed suitable for publication in PLOS ONE. Congratulations! Your manuscript is now with our production department. 

Kind regards, 

on behalf of

Dr. Wang Zhan 

Academic Editor

PLOS ONE